# Severe Acute Respiratory Syndrome Coronavirus 2 (SARS-CoV-2) Spike Protein S1 Induces Methylglyoxal-Derived Hydroimidazolone/Receptor for Advanced Glycation End Products (MG-H1/RAGE) Activation to Promote Inflammation in Human Bronchial BEAS-2B Cells

**DOI:** 10.3390/ijms241914868

**Published:** 2023-10-03

**Authors:** Dominga Manfredelli, Marilena Pariano, Claudio Costantini, Alessandro Graziani, Silvia Bozza, Luigina Romani, Paolo Puccetti, Vincenzo Nicola Talesa, Cinzia Antognelli

**Affiliations:** 1Department of Medicine and Surgery, Bioscience and Medical Embryology Division, University of Perugia, L. Severi Square, 06129 Perugia, Italy; dominga.manfredelli@studenti.unipg.it (D.M.); marilena.pariano@studenti.unipg.it (M.P.); vincenzo.talesa@unipg.it (V.N.T.); 2Department of Medicine and Surgery, Pathology Division, University of Perugia, L. Severi Square, 06129 Perugia, Italy; claudio.costantini@unipg.it (C.C.); luigina.romani@unipg.it (L.R.); 3Department of Medicine and Surgery, Microbiology and Clinical Microbiology Division, University of Perugia, L. Severi Square, 06129 Perugia, Italy; alessandro.graziani@studenti.unipg.it (A.G.); silvia.bozza@unipg.it (S.B.); 4Department of Medicine and Surgery, Pharmacology Division, University of Perugia, L. Severi Square, 06129 Perugia, Italy; paolo.puccetti@unipg.it

**Keywords:** COVID-19, methylglyoxal, methylglyoxal-derived AGEs, Nrf2, A549

## Abstract

The pathogenesis of coronavirus disease 2019 (COVID-19) is associated with a hyperinflammatory response. The mechanisms of SARS-CoV-2-induced inflammation are scantly known. Methylglyoxal (MG) is a glycolysis-derived byproduct endowed with a potent glycating action, leading to the formation of advanced glycation end products (AGEs), the main one being MG-H1. MG-H1 exerts strong pro-inflammatory effects, frequently mediated by the receptor for AGEs (RAGE). Here, we investigated the involvement of the MG-H1/RAGE axis as a potential novel mechanism in SARS-CoV-2-induced inflammation by resorting to human bronchial BEAS-2B and alveolar A549 epithelial cells, expressing different levels of the ACE2 receptor (R), exposed to SARS-CoV-2 spike protein 1 (S1). Interestingly, we found in BEAS-2B cells that do not express ACE2-R that S1 exerted a pro-inflammatory action through a novel MG-H1/RAGE-based pathway. MG-H1 levels, RAGE and IL-1β expression levels in nasopharyngeal swabs from SARS-CoV-2-positive and -negative individuals, as well as glyoxalase 1 expression, the major scavenging enzyme of MG, seem to support the results obtained in vitro. Altogether, our findings reveal a novel mechanism involved in the inflammation triggered by S1, paving the way for the study of the MG-H1/RAGE inflammatory axis in SARS-CoV-2 infection as a potential therapeutic target to mitigate COVID-19-associated pathogenic inflammation.

## 1. Introduction

COVID-19, short for “Coronavirus Disease 2019”, is a highly contagious illness caused by the novel coronavirus SARS-CoV-2.

SARS-CoV-2 infection can lead to a range of clinical manifestations, from mild respiratory symptoms to severe respiratory distress and multi-organ failure; the latter are frequently associated with an excessive and dysregulated immune response, leading to a state of hyperinflammation. The extreme release of pro-inflammatory cytokines, especially IL-1β, IL-6, IL-8, and TNF-α [1], in the context of COVID-19, is often associated with a phenomenon called the “cytokine storm”, representing a distinctive sign of the immunopathogenesis of SARS-CoV-2, in addition to being directly related to the severity and mortality of this disease. Even more critical is the fact that some COVID-19 survivors experience the so-called post-acute sequelae of COVID-19 (PASC), also known as “long COVID” or “long-haul COVID”, a range of persistent symptoms and health issues that continue for weeks or months after the acute phase of a COVID-19 infection has resolved. Even individuals who initially experienced mild cases of COVID-19 can develop these long-lasting symptoms [1,2]. It is important to note that researchers are still studying the exact mechanisms behind these long-lasting symptoms and how they relate to the initial COVID-19 infection. Some theories suggest that the virus might trigger an inflammatory response that continues even after the virus is no longer present in significant amounts in the body.

In consideration of the absence of any definitive therapeutic treatment for COVID-19, the identification of novel (and potentially selectable as targets) inflammatory pathways that can mitigate the aggressive and uncontrolled inflammatory response in COVID-19 patients could be crucial to developing effective therapeutic strategies against SARS-CoV-2 infection [3]. Currently, the exact mechanisms of SARS-CoV-2-induced inflammation are scantly known.

Methylglyoxal (MG) is a highly reactive metabolite generated as a byproduct of various cellular processes, mainly glucose metabolism [4]. It is a potent inducer of a process called glycation, occurring when MG reacts with biological macromolecules (proteins, nucleic acids, and lipids) in a non-enzymatic manner, leading to the formation of advanced glycation end products (AGEs). AGEs are stable structures that can accumulate in various tissues, a condition known as glycative stress, causing cellular damage and contributing to the development and progression of several diseases [5,6]. MG-H1 (methylglyoxal-derived hydroimidazolone 1), one of the specific products of protein glycation by MG [4], can activate inflammatory pathways, usually by binding to the receptor for AGEs (RAGE) [7,8], which is mostly expressed in the lungs, and promotes the release of pro-inflammatory molecules [8].

To counteract pro-inflammatory MG-H1-derived glycative stress, the body has a major defense mechanism, the GSH-dependent glyoxalase 1 (Glo1), that converts MG, the precursor of MG-H1, into the non-toxic D-lactate [9].

Based on these premises, in the present study, we investigated the involvement of the MG-H1/RAGE axis in SARS-CoV-2-induced pathogenic inflammation by exposing human bronchial (BEAS-2B) and alveolar (A549) epithelial cells to the SARS-CoV-2 spike 1 (S1) protein [1]. We found that S1 triggered inflammation in both cells. Interestingly, we found, specifically in BEAS-2B cells, which, unlike A549 cells, do not express ACE2-R, that S1 exerted a pro-inflammatory action through a novel MG-H1/RAGE-based pathway. Moreover, MG-H1 levels, RAGE, IL-1β and Glo1 expression levels in nasopharyngeal swabs from SARS-CoV-2-positive and -negative subjects seem to support the results obtained in vitro. Altogether, our findings reveal a novel mechanism involved in the inflammation triggered by S1, paving the way for the study of the MG-H1/RAGE inflammatory axis in SARS-CoV-2 infection as a potential therapeutic target to mitigate COVID-19-associated pathogenic inflammation.

## 2. Results

### 2.1. The Effect of the SARS-CoV-2 S1 Spike Protein on Human Bronchial BEAS-2B and Alveolar A549 Cell Viability

Before studying the pro-inflammatory effect of the SARS-CoV-2 S1 spike protein on BEAS-2B and A549 cells, viability was evaluated by an MTT assay at 6 and 24 h post-exposure at 25 and 100 ng/mL. As shown in Figure 1, SARS-CoV-2 S1 spike protein did not affect BEAS-2B and A549 viability compared to the untreated control cells. Morphological analysis, by phase-contrast imaging, confirmed the results obtained from cell viability (Appendix A). Altogether, these findings showed that at the considered doses and time course, SARS.CoV-2 S1 spike protein was not cytotoxic toward both epithelial cells.

### 2.2. SARS-CoV-2 S1 Spike Protein Induces Inflammatory Cytokines in Human Bronchial BEAS-2B and Alveolar A549 Cells

To determine the pro-inflammatory effects of S1 spike protein on BEAS-2B and A549 cells, we measured the levels of IL-1β, TNF-α, IL-6, and IL-8, crucial inflammatory cytokines in COVID-19 [1]. We found that both 25 ng/mL (Figure 2) and 100 ng/mL (Appendix A) of S1 protein induced a significant increase in all the considered cytokines compared with unexposed cells at 6 and 24 h post-exposure. Since an inflammatory status was induced in both cell models with 25 ng/mL S1 and it was more evident 24 h post-S1 treatment, we decided to perform subsequent experiments by exposing cells to these experimental conditions.

### 2.3. SARS-CoV-2 S1 Spike Protein Affects MG-H1 Levels and RAGE Expression in Human Bronchial BEAS-2B Cells

MG-H1, a major AGE derived from the spontaneous reaction of MG with the arginine residues of proteins [4], is a potent inducer of inflammatory pathways, usually through its binding to the receptor RAGE [7,8]. In the attempt to evaluate the potential involvement of both MG-H1 and RAGE in S1-induced inflammation, we detected MG-H1 levels and RAGE expression in BEAS-2B and A549 exposed to 25 ng/mL S1 for 24 h. We found that S1 induced a significant increase in MG-H1 (Figure 3a) and RAGE mRNA (Figure 3b) or protein (Figure 3c) levels in BEAS-2B cells, while it induced a modest decrease in MG-H1 levels (Figure 3d) without affecting RAGE mRNA (Figure 3e) or protein (Figure 3f) expression in A549 cells. These results suggested that the MG-H1/RAGE axis could represent a novel S1-induced pro-inflammatory pathway in human bronchial BEAS-2B cells.

### 2.4. SARS-CoV-2 S1 Spike Protein Induces Inflammation in Human Bronchial BEAS-2B Cells through MG-H1/RAGE Axis 

To demonstrate that the MG-H1/RAGE axis was involved in S1-induced inflammation in human bronchial BEAS-2B cells, we pre-treated them with 10 mM aminoguanidine (AG), which specifically reacts with MG and prevents MG-derived MG-H1 formation [6]. As shown in Figure 4, AG pre-treatment was able to completely abrogate the effects induced by S1 on RAGE expression (Figure 4a) and inflammatory cytokines (Figure 4b–e), thus confirming in BEAS-2B cells, which do not express ACE2 receptor, a novel pathway involving MG-H1/RAGE, through which S1 can orchestrate inflammation.

### 2.5. SARS-CoV-2 S1 Spike Protein Controls MG-H1/RAGE Proinflammatory Pathway through the Nuclear Factor Erythroid 2-Related Factor 2 (Nrf2)-Dependent Glo1 Downregulation in Human Bronchial BEAS-2B Cells

Intracellular MG and consequently MG-derived MG-H1 levels are controlled by the enzyme Glo1 [9], which can be upregulated by Nrf2 [10]. It is also known that Nrf2 plays anti-inflammatory roles through several mechanisms, including the suppression of pro-inflammatory genes such as IL-1β and IL-6 [11]. Based on these premises, in the attempt to understand the pathway acting upstream on the MG-H1/RAGE proinflammatory axis in BEAS-2B cells, we assumed that S1-triggered MG-H1 accumulation was dependent on Glo1 downregulation, in turn mediated by Nrf2 signaling desensitization. To demonstrate our hypothesis, we first studied Glo1 and Nrf2 expression upon exposure to 25 ng/mL S1 for 24 h. As shown in Figure 5, S1 protein was able to reduce Glo1 mRNA (Figure 3a) and protein (Figure 3b) expression and activity (Figure 5c) as well as Nrf2 signaling (Figure 5d). More importantly, pretreatment with a Nrf2 activator (Nrf2-A) [10,12], rescued Glo1 expression (Figure 5e) and activity (Figure 5f), reduced MG-H1 intracellular accumulation (Figure 5g), and RAGE expression (Figure 5h), as well as inflammation (Figure 5i,l) compared with S1-challenged cells, thus confirming that SARS-CoV-2 S1 spike protein controls MG-H1/RAGE proinflammatory pathway through Nrf2-dependent Glo1 downregulation in human bronchial BEAS-2B cells. Nrf2 deficiency is known to upregulate the ACE2 receptor, whereas the activation of Nrf2 reduces ACE2 expression, suggesting that Nrf2 activation might reduce the levels of ACE2 for SARS-CoV-2 entry into the cell [13]. In order to exclude a possible upregulation of ACE2 by the Nrf2 activator and consequently a potential involvement of ACE2 in S1-induced inflammation in our cell model, we measured ACE2 mRNA expression upon Nrf2 activator. We found no changes in ACE2 mRNA expression by the Nrf2 activator (Appendix A), thus confirming that ACE2 does not participate in S1-driven inflammation in BEAS-2B cells.

### 2.6. MG-H1 levels, RAGE, Glo1, and IL-1β Expression in Nasopharyngeal Swabs of SARS-CoV-2-Infected Patients at Different Clinical Severity of COVID-19 

To provide a potential clinical value for our in vitro data, we measured MG-H1 levels, RAGE, Glo1, and IL-1β mRNA expressions in swabs from the nasopharynx of SARS-CoV-2-infected patients, distributed on the basis of the clinical outcome into non-severe (not hospitalized: *n* = 20) and moderate/severe (total, *n* = 40: moderate, ordinary hospitalization, *n* = 20; severe, intensive care, *n* = 20) disease groups. For comparison, a SARS-CoV-2-negative group (*n* = 19) was included. The demographic characteristics of all the groups are reported in Table 1.

We found that patients with a moderate/severe disease presented higher levels of MG-H1 (Figure 6a), RAGE (Figure 6b) and IL-1β (Figure 6d), and lower levels of Glo1 (Figure 6c) expression, compared with the non-severe group, thus strongly suggesting that the mechanistic in vitro core results, showing MG-H1/RAGE pathway as a novel SARS-CoV-2 S1-driven pro-inflammatory axis, might occur during infection. The negative group presented no differences compared to the non-severe cohort with respect to the expression of all the considered genes (Figure 6).

## 3. Discussion

In the present research, we first observed, in line with the literature [1,14,15,16,17], that the SARS-CoV-2 S1 spike surface protein is sufficient alone to activate the production of key COVID-19 pro-inflammatory cytokines (IL-1β, IL-6, and IL-8) in alveolar A549 cells, a major COVID-19 cell-type target. Moreover, we also observed a comparable pro-inflammatory response in human bronchial epithelial BEAS-2B cells, which was never investigated before. Altogether, these results are important in supporting the emerging studies [1,14] that show that the S1 spike protein, regardless of the whole virus, has proinflammatory biological activity in lung epithelial cells. Despite epithelial cells usually provide feebler inflammatory responses, compared to innate immune cells, they release cytokines and/or chemokines able to potently recruit monocytes, lymphocytes, and neutrophils in the lungs infected by SARS-CoV-2, thus anyhow making a significant contribution to the “cytokine storm”-related immunopathology underpinning COVID-19 pathogenesis and progression [14]. Moreover, there is growing evidence to support that sustained and protracted circulating levels of the SARS-CoV-2 S1 spike 1 protein are present in PASC [18,19,20,21]. Our findings suggest that these plasma levels, with S1 being able to induce inflammation itself, may contribute to the sustained inflammatory-associated events, at least at the lung level, that characterize post-acute COVID-19 syndrome [22]. Hence, our data contribute to focusing on and strengthening the proinflammatory role of S1 and its implication in COVID-19 progression, including PACS, whose underlying mechanisms are entirely unknown.

Importantly, in this study, we also demonstrated that in human bronchial epithelial BEAS-2B cells, the SARS-CoV-2 S1 spike protein induces inflammation through a novel MG-H1/RAGE-based mechanism. MG-H1 is one of the specific advanced glycation end products originated by MG [4], able to initiate inflammatory pathways frequently by binding to the receptor RAGE [7,8], which is mostly expressed in the lungs [8]. As previously reported [23,24], BEAS-2B cells express very low to undetectable levels of the ACE2 receptor, to which the S1 protein typically binds in order to allow endocytosis of the viral particle and begin its replicative cycle within the host cell. Hence, our MG-H1/RAGE axis would represent a novel mechanism for S1 and potentially a novel strategy for SARS-CoV-2 to start pathogenic inflammation in cells that do not express the surface ACE2 receptor. In support of this hypothesis, in A549 cells that express ACE2 [23], MG-H1/RAGE axis is not activated by S1. 

Notably, AG, by specifically scavenging MG, thus preventing MG-derived MG-H1 formation [6], reverts RAGE-dependent inflammation. This result, in addition to demonstrating a causative role of the MG-H1/RAGE axis in driving inflammation upon S1 challenges, indicates that AG could be promising in developing supportive therapeutics to prevent COVID-19-related pathogenic inflammation. AG has already proven its potential in other inflammatory diseases [25,26]. Clearly, further investigation in this ambit is mandatory.

It is known that RAGE is a multiligand receptor [27] and that the spike glycoprotein is normally heavily glycosylated [28]. Consequently, it is also possible that the inflammatory response observed in this study may be related/potentiated via the interaction of RAGE with the carbohydrate content of the recombinant ligand tested, which cannot be excluded as a parallel event.

It is now consolidated that RAGE signaling produces a pro-inflammatory response through NF-kB activation [14,29] and, indeed, it has been recently described that SARS-CoV-2 spike protein is pro-inflammatory in pulmonary cells using NF-κB pathway activation [14,30,31,32]. Moreover, in pancreatic islets infected with SARS-CoV-2, elevated MG activates RAGE and NF-kB [33]. Finally, an upregulation of NF-kB has been observed in COVID-19 patients [34]. Hence, it is very likely that NF-kB may be involved in down-stream RAGE activation also in our model and potentially upon virus infection.

With the aim of unveiling how MG-H1 accumulation, leading to MG-H1/RAGE proinflammatory pathway activation, could occur in bronchial BEAS-2B cells, we found that this was ascribed to the downregulation, at all expression levels, of the enzyme that modulates MG intracellular levels, named Glo1. Again, the mechanism seems to be specific for ACE2-independent BEAS-2B, since ACE2-dependent A549 cells showed increased Glo1 expression (Appendix A). It has been reported that glycolysis, the main route in the production of MG, is upregulated in SARS-CoV-2-infected cells to provide the substrate required for the virus replication [35,36]. Hence, it would be possible that also this metabolic route may contribute to S1-induced MG/MG-H1 accumulation in BEAS-2B cells, even though the fact that in A549 cells where S1 should still stimulate glycolysis, MG-H1 levels do not change upon S1 treatment, which would lead one to conclude that Glo1 is indeed primarily responsible for it. 

Moreover, we found that Glo1 downregulation was dependent on S1-driven Nrf2 desensitization. It is known that the promoter region of Glo1 has a functionally operating antioxidant response element (ARE) [4] that is bound by Nrf2 to induce Glo1 expression. Here, we found that Nrf2 desensitization causatively led to Glo1 downregulation, since Nrf2 activation restored Glo1 expression and enzyme activity. In line with previous results [10], we here confirm Glo1 control by Nrf2. Interestingly, and in agreement with our findings, a very recent in vivo study shows that S1 induces Nrf2 downregulation in rats exposed to this virus subunit [37]. It is also noteworthy that a very complex interplay exists between Nrf2 and NF-kB pathways [38]. Nrf2 decreases the activation of NF-kB, while NF-kB transcription inhibits Nrf2 activation [39,40]. The existence of such a mutual interaction would seem to strengthen our hypothesis of NF-kB involvement in our pro-inflammatory Nrf2/Glo1/MG-H1/RAGE-(NF-kB) pathway, where Nrf2 desensitization could very likely explain NF-kB activation. Concomitantly, NF-kB activation could inhibit Nrf2 in a circuitry nurturing S1-driven inflammation through Glo1/MG-H1/RAGE axis, which is very intriguing and needs further investigation.

Finally, we found that patients with a moderate/severe disease presented higher levels of MG-H1 and RAGE and IL-1β expression, as well as lower levels of Glo1 expression compared with the non-severe group, thus strongly suggesting that the mechanistic in vitro core results, showing the MG-H1/RAGE pathway as a novel SARS-CoV-2 S1-driven pro-inflammatory axis, might also occur during infection. Alomar et al. have recently reported significantly higher MG and IL-1β plasma levels in patients who required intensive care unit (ICU) hospitalization compared to uninfected individuals and significantly lower Glo1 levels in patients who died compared to ICU patients that survived [41]. They also found strong inverse correlations between plasma MG and Glo1 and significant positive correlations between plasma MG and IL-1β. In the same study, the authors state, as one limitation of the research, the need for additional methodology to measure MG, Glo1, and IL-1β plasma levels. By detecting Glo1 and IL-1β levels in swabs from the nasopharynx of individuals by RT-PCR, our results appear to support those of Alomar et al.

## 4. Materials and Methods

### 4.1. Reagents

The chemicals used in this study were analytical-grade reagents from various sources. The SARS-CoV-2 S1 spike protein (cod. ab273068) was from Prodotti Gianni S.p.A (Milan, Italy) and was used at 25 and 50 ng/mL, concentrations that were not toxic in dose–response curves and were effective in inducing a pro-inflammatory response. MTT [3-(4,5-dimethylthiazol-2-yl)-2,5-diphenyltetrazolium bromide] and aminoguanidine bicarbonate (AG) were purchased from Merck Spa (Milan, Italy). Nrf2 activator (Nrf2-A, cod. 492040) was from EMD Millipore Corporation (Billerica, MA, USA). Nrf2-A was dissolved in dimethyl sulfoxide (DMSO, Merck Spa, Milan, Italy) (final DMSO concentration in incubations = 0.01%). Controls contained an identical volume of DMSO vehicle. Laemmli buffer was from Invitrogen (Milan, Italy), Roti-Block from Roth (Roth, Germany) and bicinchoninic acid (BCA) kit from Pierce (USA).

### 4.2. Cell Cultures 

Human bronchial BEAS-2B and alveolar A549 epithelial cells were from the American Type Culture Collection (ATCC). Both cells were cultured in RPMI medium supplemented with antimycotic and antibiotics (Invitrogen, Paisley, UK) at 37 °C and 5% CO_2_. 

### 4.3. Cell Viability and Morphology

Human bronchial BEAS-2B and alveolar A549 cell viability was evaluated by MTT assay [6]. Cell morphology was studied by phase-contrast microscopy.

### 4.4. RNA Isolation, Reverse Transcription, and Real-Time Reverse Transcriptase–Polymerase Chain Reaction (RT-PCR) Analyses

TRIzol Reagent was used to isolate total RNA. RevertAid™ H Minus First Strand cDNA Synthesis Kit (ThermoFisher Scientific, Milan, Italy) was used to produce cDNA, as previously described [9]. Gene expression of the studied genes versus GAPDH was evaluated by RT-PCR on a MX3000P Real-Time PCR System (Agilent Technology, Milan, Italy). The sequences of the oligonucleotide primers are reported in Table 2.

A total volume of 20 µL containing 25 ng of cDNA, 1X Brilliant II SYBR^®^ Green QPCR Master Mix, ROX Reference Dye, and 600 nM of specific primers was employed in PCR reactions, with the thermal cycling conditions being as follows: 1 cycle (95 °C, 5 min); 45 cycles (95 °C, 20 s); and (60 °C, 30 s). Comparative analysis of gene expression was performed by means of the 2−(∆∆CT) method [6].

### 4.5. TNF-α, IL-6 and IL-8 Detection 

TNF-α (cod. KHC3011), IL-6 (cod. BMS213HS), and IL-8 (cod. KHC0081) were measured using specific ELISA kits (ThermoFisher Scientific, Milan, Italy).

### 4.6. Detection of Methylglyoxal (MG)-H1 

MG-H1 was measured using ELISA, as per the manufacturer’s instructions (cat. STA-811, DBA Italia S.r.l., Milan, Italy). Fifty µL of undiluted sample was used for the assay.

### 4.7. Cell and Nuclear Lysis and Western Blot

Cell lysis was performed in radioimmunoprecipitation assay (RIPA) lysis buffer [6,9]. FractionPREP Cell Fractionation kit (Biovision, Vinci-Biochem, Florence, Italy) was used for nuclear extraction. Western blot was performed as previously described [6,9]. Samples were boiled in Laemmli buffer for 5 min, resolved on SDS-PAGE, and blotted onto a nitrocellulose membrane. Roti-Block (room temperature, RT, 1 h) was used to block unspecific binding sites. Membranes incubation with the appropriate primary Abs: mouse anti-Glo1 (D-6) mAb (dilution 1:1000, Santa Cruz, cat. # sc-133144, DBA Italia Srl, Segrate, Italy), mouse anti-RAGE (A-9) mAb (dilution 1:1000, Santa Cruz, cat. # sc-365154, DBA Italia Srl, Segrate, Italy), and mouse anti-GAPDH (6C5) mAb (dilution 1:1000, Santa Cruz, cat. # sc-32233, DBA Italia Srl, Segrate, Italy), as internal loading control, was performed overnight at 4 °C. Membranes were then incubated with the appropriate HRP-conjugated secondary Ab (RT, 1 h), and ECL was used as revealing system (Amersham Pharmacia, Milan, Italy). 

### 4.8. Nrf2 Activation Detection

Nrf2 Transcription Factor Assay kit (Colorimetric) (ab207223) (DBA Italia srl, Milan, Italy) was employed to detect Nrf2 activation in nuclear extracts, as per the manufacturer’s directions. 

### 4.9. Total Protein and Enzyme-Specific Activity of Glyoxalase 1 (Glo1) Detection

BCA kit (cat. 23225, ThermoFisher Scientific) was used to quantify total protein concentration. Bovine serum albumin was used as a standard. Glo1 enzyme activity was assayed by an established method [6]. Briefly, the assay solution contained 0.1 mol/L sodium phosphate buffer, pH 7.2, 2 mmol/L MG, and 1 mmol/L reduced glutathione (GSH). The reaction was monitored spectrophotometrically by following the increase in absorbance at 240 nm and 25 °C. One unit of activity was defined as 1 µmol of S-D lactoylglutathione produced per minute.

### 4.10. Patients

Nasopharyngeal swabs (NPS) from individual adults referred to the Hospital of Perugia for SARS-CoV-2 molecular testing were collected from December 2022 to May 2023. The study was approved by the Ethical Committee of the Umbria Region (CER Umbria, approval number 3990/21) and of the University of Perugia (15/2023 del 15 March 2023) and carried out in accordance with the Declaration of Helsinki. Informed consent was obtained from the subjects involved in the study. The cohort of SARS-CoV-2 infection cases consisted of 60 patients, stratified into 2 groups on the basis of clinical symptom severity into non-severe (asymptomatic: *n* = 20) and moderate/severe (total *n* = 40; ordinary hospitalization: *n* = 20; and intensive care: *n* = 20) groups. Once the specimens were tested for SARS-CoV-2 viral load, the residual sample was used for DNA extraction performed by using STARMag 96 × 4 Universal Cartridge Kit (Seegene, Seoul, Republic of Korea) on automated STARLET (Seegene, Republic of Korea), according to the manufacturer’s instructions. 

### 4.11. Statistical Analysis

Results were expressed as means ± standard deviation (SD) of three independent experiments. One-way analysis of variance with Dunnett’ s correction was employed to determine differences among groups. *T-test* or χ^2^ were also applied in the analysis of patients’ data where appropriate. Statistical significance was set at *p* < 0.05.

## 5. Conclusions

In summary, we demonstrated that the SARS-CoV-2 spike protein S1 induces MG-H1/RAGE activation to promote inflammation in human bronchial BEAS-2B cells in a mechanism involving Nrf2-dependent Glo1 downregulation, which is a novel finding. Our results have improved our knowledge of a better understanding of the mechanisms by which S1 and potentially SARS-CoV-2 activate inflammatory pathways, which is fundamental to finding preventive strategies and/or better treatment regimens aimed at contrasting COVID-19-related inflammation. As the specific role of the MG-H1/RAGE axis in the context of SARS-CoV-2 infection may vary depending on the emerging variants of the virus and the host’s immune response, the direct involvement of the of the MG-H1/RAGE axis in SARS-CoV-2 infection and its potential as a therapeutic target need, of course, to be further and extensively studied.

## Figures and Tables

**Figure 1 ijms-24-14868-f001:**
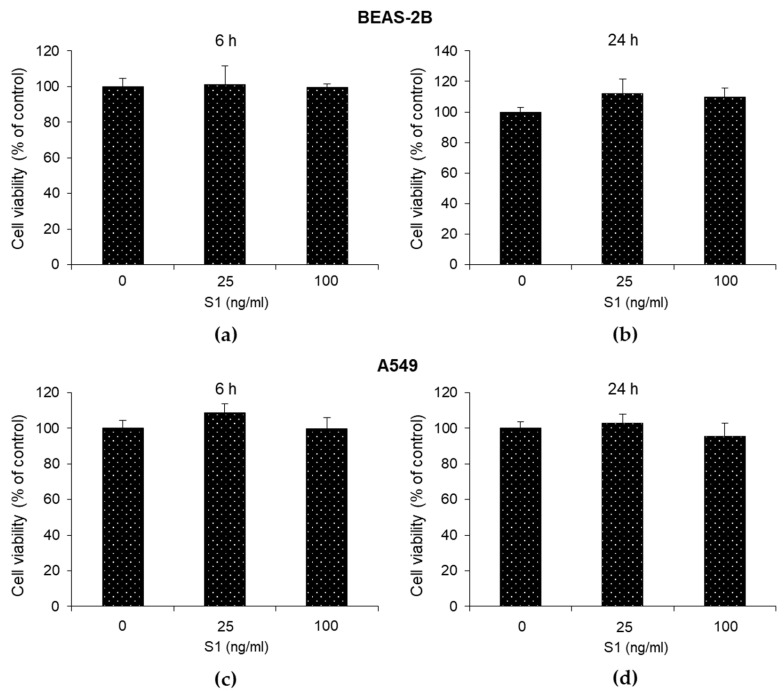
Effect of SARS-CoV-2 S1 spike protein on human bronchial BEAS-2B and alveolar A549 cell viability. Viability of (**a**,**b**) BEAS and (**c**,**d**) A549 cells, unexposed or exposed, to S1 was measured by MTT assay. Data report the means of three separate experiments carried out in triplicate, and error bars represent the standard deviation (SD) of the mean.

**Figure 2 ijms-24-14868-f002:**
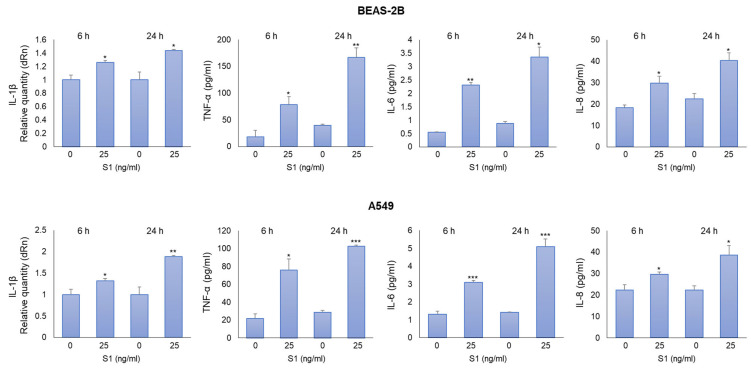
SARS-CoV-2 S1 spike protein induces inflammatory cytokines in human bronchial BEAS-2B and alveolar A549 epithelial cells. BEAS-2B and A549 cells were stimulated with S1 at a concentration of 25 ng/mL. Six and 24 h post-stimulation, the expression of IL-1β was evaluated by real-time RT-PCR, while TNF-α, IL-6, and IL-8 levels were assessed by ELISA. Data represent mean ± SD (*n* = 3). * *p* < 0.05, ** *p* < 0.01, *** *p* < 0.001.

**Figure 3 ijms-24-14868-f003:**
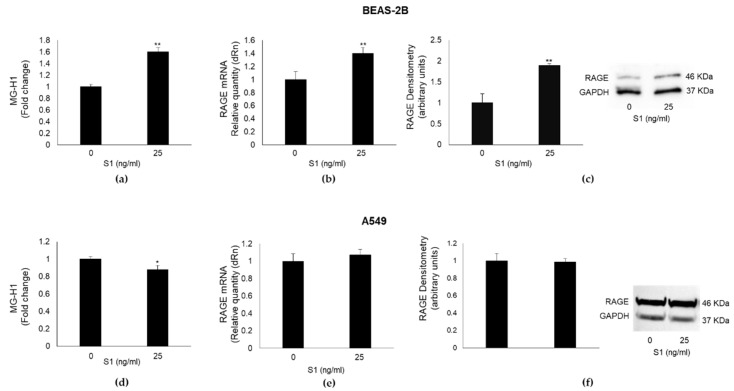
SARS-CoV-2 S1 spike protein affects MG-H1 (methylglyoxal-derived hydroimidazolone) levels and RAGE (receptor for advanced glycation end products) expression in human bronchial BEAS-2B cells. BEAS-2B and A549 cells were stimulated with S1 at a concentration of 25 ng/mL, 24 h post-stimulation, and the levels of MG-H1 (**a,d**) were evaluated by a specific ELISA kit, while RAGE mRNA (**b**,**e**) and protein expression (**c,f**) were evaluated by real-time RT-PCR and Western blot (WB), respectively. GAPDH (Glyceraldehyde-3-phosphate dehydrogenase) was used as internal loading control for WB normalization. MG-H1 absolute levels in BEAS-2B cells (mean ± SD): at 0 ng/mL S1 = 0.53 ± 0.02 µg/mL, at 25 ng/mL S1 = 0.85 ± 0.04; in A549 cells: at 0 ng/mL S1 = 0.68 ± 0.02 µg/mL, at 25 ng/mL S1 = 0.60 ± 0.03. Data represent mean ± SD (*n* = 3). * *p* < 0.05, ** *p* < 0.01. Whole blots are shown in Appendix A.

**Figure 4 ijms-24-14868-f004:**
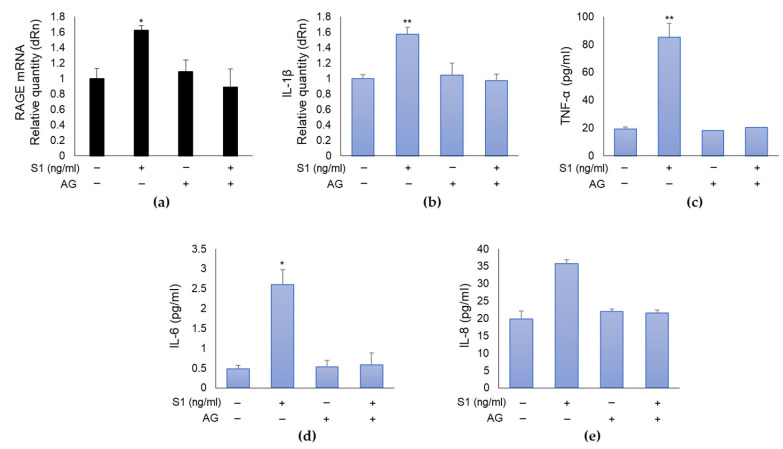
SARS-CoV-2 S1 spike protein induces inflammation in human bronchial BEAS-2B cells through MG-H1/RAGE (hydroimidazolone/receptor for advanced glycation end products) axis. BEAS-2B cells were pretreated with the specific MG (methylglyoxal) scavenger AG (aminoguanidine) (10 mM for 24 h). 25 ng/mL S1 spike protein was then added and left for further 24 h. The expression of RAGE (**a**) and IL-1β (**b**) was evaluated by real-time RT-PCR, while TNF-α (**c**), IL-6 (**d**) and IL-8 (**e**) levels were evaluated by ELISA. Data represent mean ± SD (*n* = 3). * *p* < 0.05, ** *p* < 0.01.

**Figure 5 ijms-24-14868-f005:**
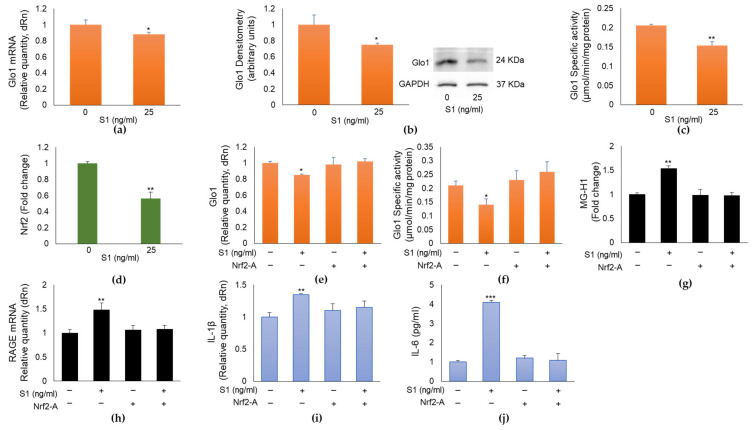
SARS-CoV-2 S1 spike protein controls MG-H1/RAGE (hydroimidazolone/receptor for advanced glycation end products) proinflammatory pathway through Nrf2-dependent glyoxalase 1 (Glo1). Glo1 downregulation in human bronchial BEAS-2B cells. BEAS-2B cells were exposed to 25 ng/mL S1 for 24 h and (**a**) Glo1 mRNA, (**b**) protein expression, (**c**) specific activity, evaluated by real-time RT-PCR, Western blot (WB) and spectrophotometric assay, respectively, and (**d**) Nrf2 nuclear expression, evaluated by a specific assay, were studied. Pretreatment with 10 µM Nrf2-A (Nrf2 activator) rescued (**e**) Glo1 expression and (**f**) activity, reduced (**g**) MG-H1 intracellular accumulation, measured by a specific ELISA kit; (**h**) RAGE expression, evaluated by real-time RT-PCR, as well as inflammation, evaluated by IL-1β (**i**) and IL-6 levels (**j**), by real-time RT-PCR and ELISA, respectively, compared with S1-challenged cells. MG-H1 absolute levels (µg/mL) in order (mean ± SD): 0.54 ± 0.012, 0.83 ± 0.022, 0.529 ± 0.044, 0.527 ± 0.023. GAPDH (Glyceraldehyde-3-phosphate dehydrogenase) was used as internal loading control for WB normalization. Data represent mean ± SD (*n* = 3). * *p* < 0.05, ** *p* < 0.01, *** *p* < 0.001. Whole blots are shown in Appendix A.

**Figure 6 ijms-24-14868-f006:**
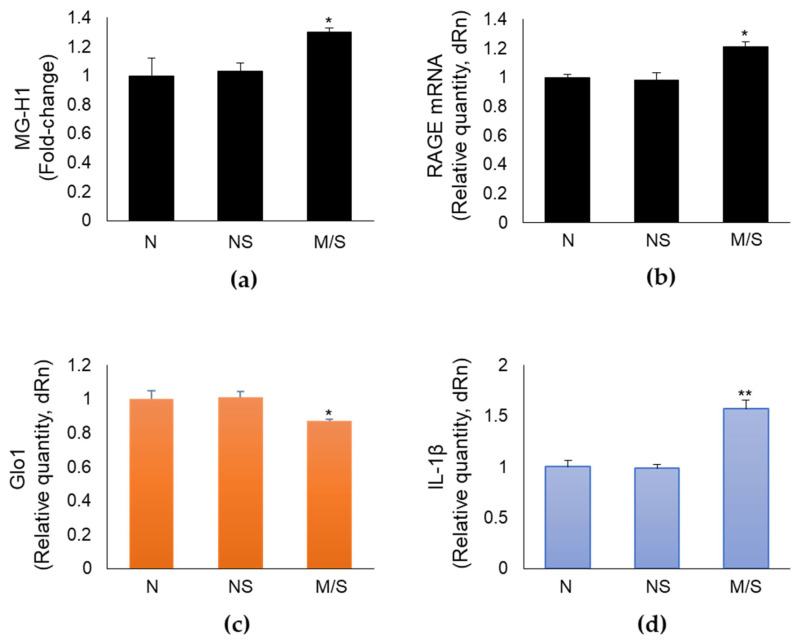
Methylglyoxal-derived hydroimidazolone (MG-H1), RAGE (receptor for advanced glycation end products), Glo1 (Glyoxalase 1), and IL-1β expression in nasopharyngeal swabs of SARS-CoV-2-infected patients at different clinical severity of COVID-19. MG-H1 levels (**a**) were measured by a specific ELISA kit (MG-H1 absolute levels (mean, µg/mL, ± SD) in the following order: 0.65 ± 0.078, 0.67 ± 0.039, 0.845 ± 0.0182); gene expression of (**b**) RAGE, (**c**) Glo1, and (**d**) IL-1β was evaluated by real-time RT-PCR. N: SARS-CoV-2-negative group; NS: SARS-CoV-2-positive with non-severe COVID-19 disease; M/S: SARS-CoV-2-positive with moderate/severe COVID-19 disease. * *p* < 0.05, ** *p* < 0.01.

**Table 1 ijms-24-14868-t001:** Demographic characteristics.

	NS	M/S	N
Age (years)			
Mean ± SD	64.75 ± 19.3	76.05 ± 16.0 *	59.10 ± 17.6 **
Median	65	80	59
Range	32–89	29–96	29–86
Gender			
Male	45% (*n* = 9)	55% (*n* = 22)	42% (*n* = 8)
Female	55% (*n* = 11)	45% (*n* = 18)	58% (*n* = 11)

NS: SARS-CoV-2-positive with non-severe (not hospitalized: n = 20) COVID-19 disease; M/S: SARS-CoV-2-positive with moderate/severe (total, n = 40: moderate, ordinary hospitalization, n = 20; severe, intensive care, n = 20) COVID-19 disease; N: a SARS-CoV-2-negative group (n = 19); SD: standard deviation; * *p* = 0.03 vs. NS; ** *p* = 0.34 vs. NS and *p* = 0.0012 vs. M/S.

**Table 2 ijms-24-14868-t002:** Sequences of oligonucleotides primers.

Gene	Sense Primer (5′-3′)	Antisense Primer (5′-3′)
Glo1	AGAAAGCACGGGGTGAAACTG	TACACCTTCAGTCCCGACTCC
IL-1β	GGACCTGGACCTCTGCCCTCTGG	GCCTGCCTGAAGCCCTTGCTGTAG
RAGE	TGAAGGAACAGACCAGGAGACAC	GCACAGGCTCCCAGACAC
GAPDH	CAAGGTCATCCATGACAACTTTG	GTCCACCACCCTGTTGCTGTAG

## Data Availability

Data are contained within the article or Appendix A.

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
