# Peer review of "Severe Acute Respiratory Syndrome Coronavirus 2 (SARS-CoV-2) Spike Protein S1 Induces Methylglyoxal-Derived Hydroimidazolone/Receptor for Advanced Glycation End Products (MG-H1/RAGE) Activation to Promote Inflammation in Human Bronchial BEAS-2B Cells"

_ijms, 2023, doi:10.3390/ijms241914868_

Round 1
Reviewer 1 Report
The manuscript submitted with the title: “SARS-CoV-2 Spike Protein S1 Induces MG-H1/RAGE Activation to Promote Inflammation in Human Bronchial BEAS-2B Cells” of Dominga Manfredelli, et al. found that SARS-CoV-2 S1 spike protein increases production of proinflammatory cytokines in both bronchial BEAS-2B and alveolar A549 cells. They also found that SARS-CoV-2 spike protein induces inflammation in the human bronchial BEAS-2B cells through activation of MG-H1/RAGE in a mechanism involving Nrf2-dependent Glo1 down-regulation. The study is original, compelling, and well-written. Hence it may be considered for publication after addressing the following comments
Major flaws
· Introduction
1. The authors should address the issue related to the post-acute sequelae of COVID-19 (PASC) linked to the cytokine storm in the pathogenesis of COVID-19, responsible for the increased MG accumulation.
2. The hypothesis to be tested should be included in the introduction
· Results
1. Authors need to justify the doses of SARS-CoV-2 S1 protein (25 & 100 ng) that are used in this study
2. It is hard to follow the results in Figure 3 because there are 6 diagrams and only 4 panels (a, b, c, and d). Thus, it is easier to have 6 panels in this figure (a, b, c, d, e, and f,) in each diagram
3. As above, it is hard to follow the results in Figure 5 because there are 10 diagrams and only 7 panels
4. The patient demographic information should be displayed in a table
· Discussion
1. The following sentence from lines 214 until 216, “Altogether, these results are important in confirming the growing body of studies” needs references
2. In line 275, “Nrf2 decreases the activation of NF-kB while NF-kB transcription inhibits Nrf2 activation [40,41]” refers to a review article rather than the original experimental work.
3. Authors concluded that the high levels of RAGE and IL-1β and levels of Glo1 in the patients with moderate/severe COVID-19 disease results from activation of MG-H1/RAGE pathway (see lines 282 to 286) without measuring MG-H1 in the nasopharyngeal swabs of SARS-CoV-2 infected patients. Thus, authors need to measure MG-H1 levels in nasopharyngeal swabs in both COVID-19 patients and the control group to arrive at this conclusion
· Methods
1. Authors should include more details about the measurement of MG-H1 levels, Nrf2 activity, and Glo-1 levels and activity such as sample dilution, incubation time, etc. to allow others to replicate their work.
2. To allow replication and ensure data accuracy, the levels of MG-H1 must be expressed as absolute numbers (µg/mL) rather than as fold changes as per the manufacturer’s instructions (cat. STA-811, DBA Italia S.r.l., Milan, Italy)
Minor
1. 13 references out of 46 are self-citation # 3, 4, 6, 9,10,11, 27, 36, 37, 43, & 44), and some of them are unnecessary e.g. 43 & 44
2. There are some typos and incomplete sentences. Please brush-up.
3. In line 74, the authors should define RAGE at the beginning
4. The following abbreviation: aminoguanidine (AG), has been defined multiple times
5. In line 293, the authors mentioned that they measured MG levels in swabs from the nasopharynx of individuals, by RT-PCR but that was not reported elsewhere in the manuscript.

Minor editing of English language required
Author Response
Dear Reviewer,
thank you for providing helpful comments and suggestions aimed at improving the manuscript and strengthen the impact of our research work.
Kind regards,

Reviewer 2 Report
Review for IJMS manuscript #2607374
Dear Sir/Madam,
I am writing to thank you very much for providing me with the opportunity to review the IJMS manuscript #2607374 entitled ‘SARS-CoV-2 spike protein S1 induces MG-H1/RAGE activation to promote inflammation in human bronchial BEAS-2B cells’.
This manuscript describes the major experimental findings of a research effort attempting to explore the potential involvement of the RAGE receptor in the induction of inflammatory responses during infection by SARS-CoV-2. A number of earlier published studies (https://doi.org/10.3390/biom11060876, https://doi.org/10.1016/j.intimp.2021.107806, https://doi.org/10.1172/jci.insight.157499) have highlighted this receptor as a potential contributor to the multidimensional inflammatory responses during the host infection by SARS-CoV-2. The approach was initially based on two in vitro cell systems, the human bronchial BEAS-2B and alveolar A549 epithelial cell lines that were stimulated with recombinant SARS-CoV-2 S1 spike protein.
Although the stimulation induced a comparable inflammatory profile in both cell lines, the BEAS-2B cells express very low to undetectable levels of the ACE2 receptor https://doi.org/10.1016/j.cell.2020.04.035, https://doi.org/10.1016/j.jcyt.2021.02.009 , which is probably the most well studied SARS-CoV-2 receptor to date (See also, TMEM106B: https://doi.org/10.1016/j.cell.2023.07.005, https://doi.org/10.1016/j.cell.2023.06.005 as well as potentially others: https://doi.org/10.1038/s41580-021-00418-x). Hence, the authors attempted to explore whether the RAGE receptor primes the inflammatory responses in BEAS-2B cells, independently of the ACE2 receptor. This is an interesting theme to study since the cell surface RAGE receptor is biochemically a multivalent receptor that has been investigated as a contributor to various disease-associated inflammatory responses (https://doi.org/10.1016/j.micinf.2004.08.004, https://doi.org/10.1186/1479-5876-7-17, https://doi.org/10.1038/srep08931, https://doi.org/10.1021/acs.jmedchem.7b00058). I am referring to the term multivalent, because although the authors focused their structured debate on the MG-H1 AGEs as a RAGE ligand in their approach, but to my understanding, they studied in vitro the biochemical stimulation of the two cell lines using recombinant S1 glycoprotein.
I read the manuscript in detail and I overall think that it is an interesting study attempting to investigate mechanistically the potential involvement of the RAGE receptor as an S1 receptor. The downstream results with BEAS-2B cells, primarily the ones from with the Nrf2 and Glo1 functional assays (Fig. 5), suggest that the RAGE receptor is probably the inducer of the inflammatory responses in these cells. In section 2.5 the term inhibit might be better replaced with reduce.
Although such responses appear comparable between the two similarly treated cell lines of the study (Fig. 2), in physiological terms, the contribution of RAGE to the induction of inflammation in patients with moderate to severe infection maybe relatively more minimal compared to the ‘canonical’ ACE2-mediated pathway (Fig. 6). This is because the levels of RAGE mRNA expression, and the Glo1 levels appear relatively close between the non-severe and moderate to severe cases of infection. One, has to take into account however, that the levels presented in Fig. 6 may also be affected by secondary effects caused by the establishment of the virus in the host during moderate to severe infection and the potential metabolic changes introduced in the metabolic bioenergetics balances in the host (feedback loop with potential up regulation of MG-H1-like derivatives).
I think that this study is interesting, however the authors need to address some important points:
-The authors discuss in the title and in text the MG-H1/RAGE axis, but in fact they are not testing any well defined advanced glycation end products (AGEs) as ligands, but instead a recombinant version of the S1 subunit of the SARS-CoV-2 glycoprotein. The spike glycoprotein is normally heavily glycosylated (https://doi.org/10.1126/science.abb9983) and the responses recorded in this study maybe related with the interaction of the RAGE receptor with the carbohydrate content of the recombinant ligand tested (If it is synthesized in commonly used hosts capable of adding post-translational carbohydrates such as HEK293T or CHO cells, insect cells etc). Therefore, I think that the authors should state the specifications of the S1 spike protein tested (Cat. number, synthesis origin, version of spike variant) in section 4.1 and perhaps consider revising some parts of the text to reflect the multivalent nature of the RAGE receptor and the mechanistic concepts discussed. Also, an updated modified title might possibly read as: ‘SARS-CoV-2 spike protein S1 induces RAGE induced inflammation in human bronchial BEAS-2B cells’.
-Although the BEAS-2B cells express very low to undetectable levels of the ACE2 receptor, it may worth mentioning in the discussion that you cannot totally rule out the possibility that the the effects you measured could be due to very low ACE2 expression or even signaling of S1 through another receptor. These cells present a slow viral replication profile (https://doi.org/10.1016/j.jcyt.2021.02.009) and ACE2 expression can be upregulated with low levels of interferon signaling (https://doi.org/10.1016/j.cell.2020.04.035). An anti-RAGE blocking antibody (https://doi.org/10.1016/j.bbi.2017.01.008) or even inhibition of RAGE with RNAi could potentially be the ultimate test of a direct S1-RAGE interaction
-Since the NRF2 deficiency (or desentization) is known to upregulate ACE2, whereas the activation of Nrf2 (with oltipraz) reduces ACE2 expression, suggests that NRF2 activation might reduce the levels of ACE2 for SARS-CoV-2 entry into the cell (https://doi.org/10.1016/j.tips.2020.07.003). If the BEAS-2B cells expressed very low to levels of the ACE2, would you expect to see results similar to Fig 5. upon Nrf2 activation? Which is the Nrf2 activator that you selected, please include the specs in section 4.1.
-There have been a few well-written reviews for long adult covid recently, for example, https://doi.org/10.3390/ijms241612962 if you wish to cite it in the introduction section.
-In addition, you may wish to include some of the refs above where necessary throughout the manuscript to strengthen your discussion.
Author Response

(The authors gave the same response as above.)

Reviewer 3 Report
Dear Author,
This is an interesting scientific paper and the topic is highly relevant. The manuscript focused on involvement of MG-H1/RAGE axis as a potential novel mechanism in SARS-CoV-2 induced inflammation using human bronchial BEAS-2B and alveolar A545 epithelial cells. However, the specific role of the MG-H1/RAGE axis in in the context of SARS-CoV-2 infection may vary depending on the emerging variants of the virus and the host´s immune response. The direct involvement of the of MG-H1/RAGE axis in SARS-CoV-2 infection and its potential as a therapeutic target needs to be extensively studied.
Below, please find suggestions for minor changes:
Overall, the structure of this manuscript appears adequate and well divided in the different sections.
KEYWORDS: COVID-19; SARS-CoV-2 spike protein 1 (S1); methylglyoxal; MG-H1; Glyoxalase 1; RAGE; inflammation; Nrf2; BEAS-2B; A549. The title words should not be repeated in ”Keywords”.
ABSTRACT
Line 20-21: Methylglyoxal (MG) is a glycolysis-derived by-product, endowed with a potent glycating action, leading to the formation of advanced glycation end products (AGEs), the major of which is main one being MG-H1.
INTRODUCTION
Line 61-64: MG-H1 (methylglyoxal-derived hydroimidazolone 1), one of the specific products of protein glycation by MG [3], can activate inflammatory pathways, usually by binding to the receptor RAGE [4,7,8], which is mostly expressed in the lungs, and promotes the release of pro-inflammatory molecules [4,8].
Line 71-74: We found that S1 triggered inflammation in both cells. Interestingly, we found, specifically in BEAS-2B cells which, unlike A549 cells,that do not express ACE2-R, that S1 exerted a pro-inflammatory action through a novel MG-H1/RAGE-based pathway.
RESULTS Please explain all abbreviations in Legends.
DISCUSSION
Line 221-223: Moreover, there is growing evidence to support that persistent systemic circulating levels of the SARS-CoV-2 S1 spike1 protein are associated with PASC [18-21].
Line 225-226: Hence, our data contribute to focus on and strengthen the proinflammatory role of S1 and its implication in COVID-19 progression, including PACS.
Line 240-241 Notably, aminoguanidine (AG), by specifically scavenging MG, thus preventing MGderived MG-H1 formation [6,10], reverted RAGE-dependent inflammation. (Please, check this past sentence)
Line 241-242 This result, in addition to demonstrating a causative role of MG-H1/RAGE axis in driving inflammation upon S1 challenge, indicates that AG could be promising in developing supportive therapeutics to prevent COVID-19-related pathogenic inflammation.
Line 252-257: Hence, it is very likely that NF-kB may be involved in down-stream RAGE activation also in our model and potentially upon virus infection. With the aim of unveiling how MG-H1 accumulation, leading to MG-H1/RAGE proinflammatory pathway activation, could occur in bronchial BEAS-2B cells, we found that this was ascribed to the down-regulation, at all expression levels, of the enzyme that modulates MG intracellular levels, named Glo1.
Line 261-264: Hence, it would be possible that also this metabolic route may contribute to S1-induced MG/MG-H1 accumulation in BEAS-2B cells, even though the fact that in A549 cells where S1 should still stimulate glycolysis, MG-H1 levels does not change upon S1 treatment, which wouldmake lead one to conclude that Glo1 is indeed primarily responsible for it.
Line 271-273: Interestingly, and in agreement with our findings, a very recent in vivo study has demonstrated shows that S1 induces Nrf2 down-regulation in rats exposed to this virus subunit [38].
Line 276-279: The existence of such a mutual interaction would seem to strengthen our hypothesis of NF-kB involvement in our pro-inflammatory Nrf2/Glo1/MG-H1/RAGE-(NF-kB) pathway, where Nrf2 desensitization could very likely explain NF-kB activation.
CONCLUSIONS: I would recommend the author to discuss the potential impact of the limitations when interpreting the study results.
Overall, the structure of this manuscript appears adequate and well divided in the different sections.
Author Response

(The authors gave the same response as above.)

Round 2
Reviewer 2 Report
Dear Sir/Madam,
I am writing to thank you very much for providing me with the opportunity to review the revised manuscript IJMS-2607374. The authors put a considerable amount of effort to improve the clarity and flow of the text and figures. In addition, they expanded the discussion concerning the potential contributions of the MG-H1/RAGE pathway to the pathophysiology of Covid-19. The discussion now includes more enhanced potential mechanistic elements, both concerning the activation of the RAGE receptor by S1, as well as the regulation of RAGE pathway at the transcriptional level by the Nrf2 and NF-kB interaction dynamics.
Overall, this is an interesting study that can expand our understanding of the multidimensional interactions of SARS-CoV-2 with the various host cell receptor signalling pathways.
The English is good. Please add the definition for PACS in page 10; Post-acute COVID syndrome (PACS).